# Synthesis and Characterization of Magnetic Drug Carriers Modified with Tb^3+^ Ions

**DOI:** 10.3390/nano12050795

**Published:** 2022-02-26

**Authors:** Dorota Nieciecka, Aleksandra Rękorajska, Dariusz Cichy, Paulina Końska, Michał Żuk, Paweł Krysiński

**Affiliations:** Faculty of Chemistry, University of Warsaw, Pasteura 1, 02-093 Warsaw, Poland; olumbalum@gmail.com (A.R.); dariusz.cichy@student.uw.edu.pl (D.C.); p.konska@student.uw.edu.pl (P.K.); mt_zuk@chem.uw.edu.pl (M.Ż.); pakrys@chem.uw.edu.pl (P.K.)

**Keywords:** magnetic nanoparticles, drug carriers, magnetic hyperthermia, Langmuir monolayers

## Abstract

The study aimed to synthesize and characterize the magnetic drug carrier modified with terbium (III) ions. The addition of terbium extends the possibilities of their applications for targeted anticancer radiotherapy as well as for imaging techniques using radioisotopes emitting β^+^, β^−^, α, and γ radiation. The synthesis of iron oxide nanoparticles stabilized with citrates using the co-precipitation method (IONP @ CA) was carried out during the experimental work. The obtained nanoparticles were used to synthesize a conjugate containing terbium ions and guanosine-5′-monophosphate as an analog of drugs from the thiopurine group. Conjugates and their components were characterized using Transmission Electron Microscopy, infrared spectroscopy, X-ray microanalysis, spectrofluorimetry, and thermogravimetric analysis. The hybrid was also investigated with Langmuir layers to check the interaction with analogs of biological membranes.

## 1. Introduction

A lot of scientific attention is nowadays paid to magnetic nanoparticles (MNP) [1,2]. Although many materials can form magnetic nanoparticles—pure metals, metal alloys, or metal oxides—iron oxides seem to play the most significant role. Iron oxide nanoparticles (IONPs) are built of magnetite (Fe_3_O_4_) or maghemite (Fe_2_O_3_), or—more often—of non-stoichiometric iron oxides [3]. Although for years, they were believed to be biocompatible and low toxic [4], it turned out that toxicity needs further studies and depends not only on the way of administration but also on coating, dosage, and hydrodynamic diameter [5,6].

The unique magnetic properties of these objects—high values of magnetization related to the unpaired electrons of iron cations, and superparamagnetism phenomenon occurring for nanoparticles smaller than 30 nm [3,7]—make them the ideal candidates for medical applications [8]. IONPs can be used simultaneously in therapies and diagnostics of different types of cancers [9], making them particularly suitable for theranostics [10]. Numerous studies prove that they can be used as drug carriers [11,12,13], as contrast agents in magnetic resonance imaging (MRI) to diagnose abnormalities [14], to control over the drug release process [8,15], or as localized heating sources in magnetic hyperthermia, initiating the cancerous tissue self-disruption [16,17]. Not all IONPs possess all of the properties mentioned above, and efforts are made to design the optimal synthetic route that will lead to the desired product [2].

The IONPs’ surface can be relatively easily engineered, and as a result, multi-functional structures are formed [15,18,19,20]. The surface coating is vital for this type of MNP because of their intrinsic instability–susceptibility to corrosion leading to structural changes and magnetization loss. Usually, such a coating layer on the surface additionally provides stability in biological media and may improve the cellular uptake, tissue distribution, and clearance, reducing also IONPs’ toxicity [15,21,22]. Moreover, the coating layer on nanoparticles can be designed to increase their affinity to drugs, making IONPs suitable as magnetically targeted drug delivery platform in various therapies, limiting the drug distribution solely to the e.g., tumor tissue, and therefore reducing the drug amount necessary for use as compared with classical intra- or extravenous administration. This is of great importance because the adverse effects of conventional modes of treatment often are decisive in their premature end. A very good example of such an approach can be found in nuclear medicine [23,24,25], in which the immobilizing of radioisotopes within/on the magnetic structure allows the so-called endoradiotherapy, limiting the area of radiation solely to the tumor and minimizing damage to healthy tissue.

In this study, we present an advanced magnetic conjugate composed of IONPs covered with a layer of citric acid (CA) with guanosine-5′-monophosphate (GMP) and Tb^3+^ attached to it. Guanine in the GMP can be easily replaced with a drug from the mercaptopurine group, such as 6-thioguanine or 6-mercaptopurine. Compared to the guanine, these compounds (thiopurines in general) have an oxygen atom substituted with a sulfur atom in the ring structure, and such replacement is responsible for the development of the antitumor properties of thiopurines. The antitumor properties of the conjugate are related to the presence of the GMP compound, in which guanine is replaced with 6-thioguanine or 6-mercaptopurine. The first clinical application of thiopurines (TP) in the treatment of leukemia was published by Burchenal et al. in 1953 (especially acute lymphocytic, acute myeloid, and chronic myeloid leukemia) [26]. Currently, analogs of purine bases are used not only in the treatment of acute forms of leukemia. In recent years, the interest in mercaptopurines has clearly increased due to the promising results in the treatment of prostate cancer and glioma, especially in patients who failed conventional chemoradiotherapy [27].

Here, we report on a potential therapeutic conjugate containing stable terbium isotope, apart from GMP. As in the case of guanine that can be replaced with thiopurine, a stable Tb isotope can be replaced with ^161^Tb emitting low-energy β^−^ radiation [28] and be used in targeted endoradiotherapy. The treatment of tumors might be more effective with low energy *β* emitters, which combine short-range and high linear energy transfer—a combination that results in the high relative biological effect and cytotoxicity [29]. The choice of ^161^Tb was based upon its half-life (^161^Tb (*t*_1/2_ = 6.9 days)), which is a time-span sufficient for the generation of the radioisotope “on-demand” and its complexation with pre-synthesized IONP-GMP system for the final conjugate.

The multiplied therapeutic action of the proposed conjugate is expected to result from (i) the effect of β^−^ radiation of ^161^Tb limited only to the distance of several millimeters, (ii) the local magnetic hyperthermia–heating of cancer cells to the temperatures above 42 °C, induced by the external alternating magnetic field and the presence of thiopurine in the final radioconjugate that can be synthesized in the specialized laboratories.

It is a promising multifunctional system that can be used in biological imaging and can be precisely guided to cancer sites.

## 2. Materials and Methods

### 2.1. Chemicals

Iron (III) chloride hexahydrate FeCl_3_·6H_2_O (97%) and iron (II) chloride tetrahydrate FeCl_2_·4H_2_O puriss p.a. *≥* 99%, terbium (III) chloride hexahydrate TbCl_3_·6H_2_O (99.9% trace metals) were obtained from Sigma-Aldrich (St. Louis, MO, USA), 25% ammonia solution NH_4_OH was supplied from POCH (Gliwice, Poland). Deionized water with resistivity 18.2 MΩ cm at 25 °C was obtained using the Milli-Q ultra-pure water filtering system from Merck (Warszawa, Poland). Citric acid was purchased from Sigma-Aldrich with 94% grade acid. Acetone having analytical grade was supplied from POCH (Gliwice, Poland).

Lipids: DOPC (1,2-dioleoyl-sn-glycero-3-phosphocholine, purity > 99%) cardiolipin (Heart, Bovine extract, purity > 99%) were purchased from Avanti Polar Lipids (Alabaster, AL, USA). Octadecylamine (>99%), guanosine 5′-monophosphate disodium salt hydrate (GMP) sodium dihydrogen phosphate, and sodium hydrogen phosphate were purchased from Sigma-Aldrich. Phosphate buffer solutions at pH 7 and pH 5 were prepared by mixing solutions of NaH_2_PO_4_ and Na_2_HPO_4._ To achieve pH 7, the solutions of NaH_2_PO_4_ and Na_2_HPO_4_ were mixed together with the final concentration 38.96 mM and 61.04 mM, respectively. In the case of pH 5, the same solutions were used with the concentration of 98.58 mM and 1.42 mM (NaH_2_PO_4_ and Na_2_HPO_4_).

### 2.2. Synthesis and Modification of Nanoparticles

The synthesis of superparamagnetic iron oxide nanoparticles was performed using the co-precipitation method adapted from Miola et al. [17]. Briefly, 0.75 g of FeCl_2_·4H_2_O and 1.75 g Fe(NO_3_)_3_·6H_2_O were dissolved in 37.5 mL and 50 mL of distilled water, respectively. Both solutions were magnetically mixed for 30 min, and the pH in both beakers was equal to 2. After mixing both solutions, the final pH was about 1.9. The nanoparticle suspension was obtained by the dropwise addition of an aqueous ammonia solution (ca 20 min, 5.5 mL) until the pH was about 10. The suspension was continuously stirred with the mechanical stirrer (500 rpm). After the ammonia addition, the solution turned black due to the nanoparticles of iron oxide formation. Then, the beaker was placed on a magnet to accelerate the decantation of the suspension. The supernatant solution was decanted, and the precipitate was rinsed three times with a mixture of H_2_O/NH_4_OH (volume ratio 1:1) to remove unreacted substrates. Then, the NPs were suspended in 100 mL of distilled water.

In this work, IONPs were modified with citric acid, also working as a drug linker with nanoparticles. As-synthesized IONPs were decanted with a magnet, and 120 mL of 0.05 M citric acid was added. In the next step, the pH was adjusted to the value 5.2 by the addition of aqueous ammonia. The mixture was heated at 90 °C and magnetically stirred at 250 rpm for 90 min to adsorb citric acid on the surface of nanoparticles. Modified IONPs were precipitated using acetone and separated by a magnet. Then, IONPs@CA was rinsed four times with distilled water; between each washing, the suspension was sonicated. Rinsed nanoparticles were suspended in a few ml of water and dried at 50 °C for 12 h.

As a result of the synthesis, about 750 mg of IONP@CA were obtained.

### 2.3. Synthesis of Conjugate IONPs@CA_GMP_Tb^3+^

The next step was the formation of the conjugate with the drug within the following procedure. Initially, a HEPES buffer solution of 10 mL having 0.1 M concentration and pH 7.4 was prepared (0.24 g of HEPES was weighed and dissolved in 10 mL of distilled water). In the following step, 0.1 g of guanosine-5′-monophosphate was added, and the solution was stirred for 10 min. A magnetic nanoparticle suspension was prepared in a second beaker by adding IONPs@CA (20 mg) to 10 mL of water. Both solutions were mixed, and the obtained mixture was incubated in a water bath at 25 °C for 30 min.

In the last stage, 0.13 g of Tb(NO_3_)_3_·6H_2_O was dissolved in 10 mL of distilled water and added to the previously prepared IONPs@CA_GMP solution, where the beaker was placed on a stirrer for 1.5 h. As an effect, a brown precipitate was obtained, which was centrifuged and washed four times with distilled water. This procedure was to remove excess terbium salt and guanosine 5′-monophosphate.

### 2.4. Techniques

The size of nanoparticles was investigated using Transmission Electron Microscopy (TEM) (Zeiss Libra 120 Plus, Stuttgart, Germany) and Dynamic Light Scattering (DLS) (Malvern Instruments Zetasizer Nano ZS, Malvern, UK). A drop of sample water suspension was air-dried on a Cu mesh.

The modification of the IONPs surface was characterized by FTIR spectroscopy with a Nicolet 8700 Spectrometer Fisher Scientific and zeta potential (Malvern Instruments Zetasizer Nano ZS, Malvern, UK). IR spectra were measured in KBr pellets. Thermogravimetric analysis (TGA) was performed with TGA Q50 (TA Instruments, New Castle, PA, USA). Approximately 10 mg of dry sample was placed on a platinum pan. The measurement was carried out in an inert gas (nitrogen), and the sample was heated with a rate of 10 °C/min in the range of 25 to 800 °C.

Magnetic characterization of powdered samples was performed employing a vibrating sample magnetometer (VSM, Quantum Design, San Diego, CA, USA). The hysteresis loops were recorded for a few milligrams of dried, uncoated IONPs placed in a disposable plastic holder with the range of magnetic field −2.0 and +2.0 T at a temperature of 300 K.

Fluorescence spectroscopy was used to investigate the fluorescent properties of terbium coat onto the IONPs. First, 3 mL of diluted conjugate suspension was poured into a quartz cuvette and then placed in spectrofluorometer, Fluorolog 3-2-IHR320 (Horiba Jobin Yvon, Lier, Belgium). The excitation wavelength was 325 nm, which resulted in an IONPs@CA_GMP_Tb3+ fluorescence spectrum ranging from 450 to 600 nm. The UV-vis absorption spectra of released GMP from conjugate were obtained using a Perkin Elmer Lambda 35 spectrometer (Waltham, MA, USA). Then, 2 mg of the conjugate was suspended in a volume (5 mL) of phosphate buffer saline (PBS) at various pH values (pH 5 and pH 7) at room temperature. The resulting suspension was placed in a vial for 7 h, and 100 μL were taken out of the solvent at appropriate time intervals and replaced by the same volume of fresh PBS buffer to keep the total volume of the release medium constant. Before each sampling (100 µL), the conjugate was precipitated on the permanent neodymium magnet. Then, 100 μL of supernatant was diluted with water to 2 mL in a cuvette.

The magnetic hyperthermia (MH) experiments were performed with nanoscale Biomagnetics D5 Series equipment (Zaragoza, Spain) with CAL1 CoilSet. The Specific Absorption Rate (SAR) values were estimated using MaNIaC Controller software with ZaR subprogram (nB nanoScale Biomagnetics, Zaragoza, Spain). The magnetic properties of samples were verified with a QD vibrating sample magnetometer VSM over the magnetic field range from −2.0 T to +2.0 T at a temperature ranging from 100 to 300 K with an accuracy of ca. 0.01 K.

Biomimetic membranes were formed with a KSV-Nima KN2003 trough system with the KSV NIMA LB Software (Biolin Scientific, Manchester, UK). The Langmuir trough was cleaned with chloroform and methanol before each measurement. Next, the thoroughly cleaned Langmuir trough was filled with the subphase solution: milli-Q water or the suspension of IONP nanoparticles or their conjugates. Then, 30–40 µL of a chloroform solution of the selected lipid at a concentration of 2 mg/mL was applied onto the surface of the subphase. After about 20 min, when the solvent had evaporated, the lipid layer was compressed. The experiments were carried out with a 5 cm^2^/min barrier compression rate until the surface pressure reached the value of 30 mN/m. This value corresponds to the pressure naturally occurring in living cells. In order to attain the equilibrium state, usually, 20 min was sufficient both to evaporate the solvent from the surface of a subphase, but additionally, several subsequent compression–decompression isotherms were recorded at time intervals of 20 min to confirm the stability of the isotherm shape. No differences were observed for the same lipid monolayer and the same nanoparticle suspension (Appendix A).

Cytotoxicity studies were performed by MTS colorimetric assay in the concentration range of conjugates 0–100 µg/mL. MDA-MB-231 cells were seeded 24 h before the experiment in 96-well plates at a density of 3.0 × 10^3^ per well. Then, the cells were washed with PBS and treated with increasing concentrations of the studied conjugates to a volume of 100 µL. Seeded cells were incubated with nanoparticles for 18 h, washed with 100 µL of PBS, and incubated for another 24, 48, and 72 h at 37 °C. Next, the MTS assay was added to each well, and plates were incubated for an additional 2 h at 37 °C in the dark. Lastly, the absorbance was measured at 492 nm using the Apollo 11LB913 microplate reader, Berthold (Bad Wildbad, Germany) (Appendix A).

## 3. Results

The formation and physicochemical characterization of the conjugates was performed at each step of their synthesis.

### 3.1. Characterization of IONPs@CA_GMP_Tb^3+^

#### 3.1.1. Morphology Analysis

The morphology of the nanoparticles was examined using a Transmission Electron Microscope to determine their shape, size, and level of dispersion. The TEM images of the IONPs@CA (Figure 1) and the IONPs@CA_GMP_Tb^3+^ (Figure 2) were taken for particles that were initially dispersed in water and then dried on a Cu mesh. TEM image shows that the nanoparticles with citric acid have a uniform, well-defined, and almost spherical shape. Little agglomeration is observed here, which was most probably due to the preparation technique of the sample. The particles’ mean diameter and standard deviation were determined from the TEM image from the average of about 260 particles and presented in the histogram. The size of the magnetic nanoparticles is between 8 and 17 nm. The average diameter of the MNP was 13 nm (±1 nm). The organic shell is too small to be clearly recognized with the TEM method.

The obtained TEM images for IONP@CA (Figure 1a) and IONP@CA_GMP_Tb^3+^ (Figure 2a) differ both in the size of the nanoparticles and their distribution. The organic layer of GMP is visible as grayish “corona” around darker cores (Figure 2a). Organic compounds have a different density than the magnetic core, so the brightness of the coat is different on the TEM image than that of the magnetic core. The conjugate shows a high tendency to agglomerate; large clusters of ligand molecules with magnetic nanoparticles are formed (Figure 2b). After the coordination of Tb^3^+ ions, we performed also the elemental EDS analysis to confirm the presence of terbium in the conjugate. The recorded spectrum shows the presence of peaks from iron, terbium, phosphorus, carbon, nitrogen, and oxygen (visible copper signal comes from the copper mesh used as a substrate in TEM experiments). EDS analysis confirmed the presence of the elements characteristic to the conjugate, confirming the successful preparation of IONPs@CA_GMP_Tb^3 +^ product (Figure 2c).

Complementary to TEM analysis, the size of nanoparticles and their distribution in an aqueous suspension were also measured in dynamic light scattering (DLS) experiments. Figure 3 shows the histogram of the nanoparticles stabilized with citric acid. One can see that nanoparticles with a size of ca. 32 nm are the most abundant. The observed larger size of nanoparticles reported in the DLS experiments as compared with TEM data (13 nm) results from the presence of the solvation shell around the hydrophilic IONP@CA nanoparticles, significantly enlarging their hydrodynamic size.

Compared to the above, the conjugates indicate a larger particles size of about (111.0 ± 4.5) nm. It may be related to the slight agglomeration of nanostructures in an aqueous solution and the possibility of creating larger structures due to the presence of an organic shell. The polydispersion index is 0.22, below 0.30, indicating that the nanocarriers exhibit a narrow particle size distribution and good dispersity.

For medical applications of our conjugates, their stability in aqueous media as the suspension is of utmost importance. Since all of the constituents of final constructs are charged, we monitored the changes of IONPs’ zeta potential as a measure of suspension stability: the higher the absolute value of zeta potential of nanoparticles, the more stable their suspension, due to the electrostatic repulsion. The highest negative value of the potential for citrate-coated nanostructures was about −35 mV, indicating that citrates stabilize the suspension, protecting against sedimentation/aggregation in the solution. During the next steps of synthesis, a complex is formed between the phosphate groups present in the GMP and the citrate ions present on the surface of the nanoparticles, which increases the value of the zeta potential to about (−15.8 ± 1.0) mV for the conjugate.

#### 3.1.2. Organic Shell Content in Conjugate—Thermogravimetric Analysis

The amount of the organic shell on the surface of nanoparticles was evaluated with thermogravimetry. Based on the graph (Figure 4), it can be concluded that citric acid constitutes 10% of the sample mass.

By comparison of the curve that was recorded for citrate-modified nanoparticles and the curve for the conjugate, it can be concluded that in the latter case, the organic compounds comprise ca. 50% to 55% of the conjugate mass. Therefore, the TGA results confirm the successful conjugation of GMP with IONPs@CA.

We have performed also the preliminary stability studies of the obtained conjugates. For this purpose, we carried out the GMP release experiments at two pH values, corresponding to the normal tissue (pH 7) and tumor tissue (pH 5) conditions (Appendix A). The release was controlled with UV-Vis spectrometry, monitoring the absorption spectrum increase in the range of 200–500 nm. At pH 7, almost no release was observed, whereas a rapid release within the first hour was observed at pH 5. These preliminary results confirm the usefulness of our conjugates in treating tumor tissues. However, further studies are necessary to quantitatively describe the release kinetics under various conditions.

To investigate the cytotoxicity of the synthesized conjugate, preliminary studies were carried out using the MTS test for the MDA-MB-231 cell line. The results are presented as metabolic activity (%) in comparison to the control not treated with the studied compounds (Appendix A).

#### 3.1.3. Magnetic Properties of Nanoparticles

Figure 5 presents the recorded magnetization hysteresis curves for nanoparticles covered with citric acid as well as IONPs@CA_GMP_Tb^3+^ conjugate. One can see that the presence of organic shell causes a slight decrease in magnetic properties as compared with IONP@CA. In addition, the magnetization of conjugates does not attain saturation. This behavior is typical for a magnetic core covered with a diamagnetic organic layer [30,31].

Our results prove that nanoparticles exhibit excellent magnetic properties. A magnetic field of 2 T is sufficient to saturate the sample, and the value of saturation magnetization (Ms) at 300 K is high and equal to 63 emu/g. This value is close to the values presented by Goya et al. [32]. They reported slightly lower Ms, equal to 60.1 emu/g, for nanoparticles with an average diameter of 11.5 nm. This is due to the diameter difference—smaller nanoparticles have a higher ratio of surface (magnetically disordered) to volume (ordered) spins, resulting in smaller Ms values [33]. Under the same conditions, the synthesized IONPs@CA_GMP_Tb^3+^ conjugates show the magnetization value of ca. 46 emu/g. This value, even though smaller, is sufficient for the magnetic hyperthermia applications, as will be discussed later in the text.

As was described in the previous section, the average nanoparticles diameter is smaller than the critical diameter for this type of material [3], and magnification of the curve at close to 0 T magnetic field strengths proves that the value of residual magnetization is close to 0 emu/g. This suggests that nanoparticles are superparamagnetic, which is very desirable from the medical applications’ point of view.

The inset in Figure 5 shows the blow-up of magnetization of IONPs at a very low external magnetic field (±0.02 T). The magnetic remanence (at zero magnetic field), M_R_, is as low as up to 2 emu/g at RT. The coercive field, H_C_, observed on the measured magnetization isotherm is quite low as well, reaching the value of 0.001 T (10 Oe), which is characteristic for the superparamagnetic material. This very low value of H_C_ is very advantageous from the point of view of possible medical applications of such nanoparticles. The low values of magnetic remanence hinder the aggregation and precipitation of nanoferrites and therefore, the hazardous blocking of vascular flow becomes reduced.

#### 3.1.4. FT-IR Studies

The following measurements were performed using FTIR spectroscopy that investigates the chemical composition of the conjugate. As can be seen in Figure 6, black spectrum, iron oxide nanoparticles are characterized by two intense peaks of stretching vibrations (580 cm^−1^ and 630 cm^−1^) related to Fe-O bonds in the IONPs crystal lattice. They are characteristic of all spinel structures, especially ferrites [34]. The two bands at 1627 cm^−1^ and 1384 cm^−1^ can be assigned to the citrate modification of nanoparticles. For comparison, the blue curve in Figure 6 was recorded for pure citric acid. Since after the synthesis, nanoparticles have been thoroughly washed to remove an excess of non-adsorbed citrates, no bands of free citric acid are expected on the FTIR spectra of IONP@CA. So, the peak at the wavenumber of 1756 cm^−1^ characteristic for the C = O stretching vibration from the carboxyl group of free acid (blue curve) disappears after binding to the surface of nanoparticles. Instead, due to the interaction with the IONP’s surface, it takes on a partial character of a single bond shifting toward the lower wavenumbers (ca. 1627 cm^−1^) and merging with dissociated COO^−^ vibrations. Additionally, the band at 1384 cm^−1^ (blue and black curves) corresponds to symmetric stretching of the COO^−^ bond. Based on these data, it can be concluded that citric acid binds to the surface of magnetic nanoparticles by the strong adsorption of citrate ions.

The IONPs@ CA_GMP_Tb^3+^ spectrum indicates the same characteristic features for IONPs and IONPs@CA nanoparticles. In addition to these, there are intense bands from the P-O stretching vibrations (1116, 1001 cm^−1^), which confirm the presence of phosphates in the conjugate. The broad bands between 3343 and 3209 cm^−1^ are characteristic of the stretching vibrations of the N-H bonds in the purine ring. They overlap with the bands 3300–3500 cm^−1^ (present in all samples) that can be assigned to the stretching vibrations of hydroxyl group O-H due to water adsorption during the experiment. The IR analysis confirms that GMP molecules were successfully bound to the nanoparticles.

#### 3.1.5. Spectrofluorimetric Studies

Next, fluorescence spectroscopy was used to investigate the fluorescent properties of terbium coat onto the IONPs.

Figure 7 shows the fluorescence spectrum for IONPs@CA_GMP_Tb^3+^ in distilled water at an excitation wavelength of 325 nm. This spectrum is characterized by emission bands at 488 nm, 545 nm, and 585 nm. They result from Tb emission transitions from state ^5^D_4_ to ^7^F_6_, ^7^F_5_, and ^7^F_4_. The obtained emission bands are narrow and intense, which proves that complex terbium ions were obtained [35].

### 3.2. Interaction of IONPs@CA and Conjugate with Biomimetic Membranes

The biomimetic membranes prepared during the experimental work are the equivalent of membranes found in the cells of living organisms. The Langmuir technique enables the formation of such analogs that are more easily controllable in composition than the natural membranes.

Therefore, in the following experiments, we present the effect of conjugates on the organization of Langmuir lipid monolayer films.

Two types of lipids and their analog—DOPC (1,2-dioleoyl-sn-glycero-3-phosphocholine), cardiolipin (heart, bovine extract), and octadecylamine, respectively—were used to form the Langmuir monolayers. The choice of film-forming substances was deliberate, because of the differences in the acid–base properties of their hydrophilic head groups as well as the structure of fatty acid chains. In the first stage, the influence of Fe_3_O_4_ iron oxide nanoparticles coated with citric acid on the formed biomimetic layers was investigated. The pressure vs. area per single lipid molecule graphs, called pressure–area isotherms (π-A isotherms), are shown in Figure 8.

The shape of all π–A isotherms obtained for the DOPC lipid in the absence and presence of nanoparticles stabilized with citrates is similar. A systematic increase in surface pressure is observed during the compression of the monolayer. The addition of Fe_3_O_4_ iron oxide nanoparticles in the concentration range of 1 mg/mL up to 10 mg/mL does not affect the surface area per one molecule of lipid. To obtain a more detailed description of the physical state and organization of the monolayer, the parameter describing the mono-layer compressibility (*C_s_*) is defined, which describes the slope of the Langmuir isotherm by Equation (1).
(1)$$C_{s}=-\frac{1}{A}{\left(\frac{dA}{d\pi }\right)}_{T,p}$$

A more commonly used term is the compressibility coefficient, being the reciprocal of the compressibility, *C_s_*^−1^. Based on our experimental data, the range of the compressibility coefficient values corresponding to the individual physical properties of the monolayer were determined, reflecting the influence/interaction of IONP@CA nanoparticles with DOPC (Figure 8a), cardiolipin (Figure 8c), and octadecylamine (Figure 8e). At the same time, Figure 8b,d,e show the compressibility coefficient plots of all three monolayer-forming molecules vs. the surface pressure within the monolayer. This allows determining the maximum value of the compressibility coefficients reflecting the degree of ordering of molecules in the monolayer and its physical state at the water/air interface following the Davies and Rideal criteria [36,37]. So, in the case of the DOPC monolayer without the IONP@CA nanoparticles, the *C_s_*^−1^ value was up to 81 mN/m at the highest surface pressure, whereas for the concentration of IONP@CA of 1 mg/mL and 10 mg/mL, the values reached 83 mN/m and 104 mN/m, respectively. The obtained values correspond to the liquid condensed phase of DOPC monolayers, and their range suggests very weak interactions of citrate-stabilized iron oxide nanoparticles on the DOPC monolayer.

Similar studies were performed for the cardiolipin. It is clearly seen that the presence of iron oxide nanoparticles stabilized with citric acid with concentrations of 1 and 10 mg/L caused only very small changes in the surface area per cardiolipid molecules at 30 mN/m surface pressure.

The presence of nanoparticles in the subphase does not cause significant changes in the structure/organization of the obtained biomimetic DOPC and cardiolipin monolayers within the whole range of surface pressures and concentrations of IONP@CA studied (Figure 8a–d). For each monolayer, *C_s_*^−1^ is about 80 to 104 mN/m, indicating that all monitored monolayers were characterized by a similar molecule ordering at the water/air interface [38].

Langmuir isotherms obtained for octadecylamine (C_18_NH_2_) film without and with Fe_3_O_4_ differ in their response to the presence of IONP@CA nanoparticles as compared to DOPC and cardiolipin monolayers. First of all, C_18_NH_2_ molecules on pristine aqueous subphase show higher ordering compared to lipids. This is due to the presence of only one saturated hydrocarbon chain (C_18_) and a small hydrophilic amino group compared to the two unsaturated fatty acid chains and large hydrophilic groups of the studied lipids. This is reflected also by the compressibility coefficient of C_18_NH_2_ monolayers, being in the range of liquid-condensed state and reaching the value of ca. 130 mN/m at 25 Å^2^ area per octadecylamine molecule. As a result of this, we expected a different effect of citrate-stabilized nanoparticles on such monolayers. However, the isotherm shapes in the absence and presence of IONP@CA nanoparticles in the subphase at a concentration of 1 mg/L were similar. In addition, the compressibility coefficients under these conditions were similar, showing that the nanoparticles at this concentration affect only negligibly the organization of the monolayer. However, a significant difference in the shape and position vs. area per molecule (X-axis) of the Langmuir isotherm was observed in the presence of 10 mg/L of nanostructures in the subphase. It was not possible to reach the surface pressure value of 30 mN/m as in the previous cases, because at ca. 26 mN/m, there was a collapse in the organization of the monolayer molecules. Further compression of the film is not possible and may remove C_18_NH_2_ molecules or nanoparticles from the monolayer structure or just form multilayers of octadecylamine/nanoparticles aggregates. Moreover, the pressure in the monolayers starts to build up at a different, larger C_18_NH_2_ area per molecule, suggesting again the incorporation and accumulation of IONP@CA nanostructures within the headgroup region of the monolayer. It can be related to the electrostatic interaction between the carboxyl groups on the surface of nanoparticles and the amino groups in the membrane structure.

The compressibility coefficients of the obtained layers (C_18_NH_2_) for the monolayer without and with the lower concentration of IONP@CA (1 mg/L) were of ca 126 mN/m. However, for the same monolayer with nanoparticles at a higher concentration (10 mg/L), this value was reduced to about 45 mN/m. It turns out that the higher amount of nanoparticles in the subphase causes poorer packing of the molecule in the obtained layers. The compressibility coefficients for the layer without the presence of nanostructures and with their low concentration (1 mg/L) proved that the condensed liquid phase was obtained. However, after adding larger amounts of nanoparticles, i.e., 10 mg/L, the obtained layer changes the phase state into an expanded liquid. Based on this information, we are inclined to speculate that the IONP@CA nanoparticles tend to slightly modify the lipid head groups to adapt less tilted orientation, particularly in the case of 10 mg/L of the nanoparticles in the subphase interacting with cardiolipin Langmuir films. However, different behavior can be observed in the case of C_18_NH_2_ monolayers and 10 mg/mL concentration of citrate-stabilized IONPs. In this case, the strong interaction of amine groups of octadecylamine and IONP@CA can be observed, resulting in a disintegrating effect exerted by the nanoparticles on the hydrophilic part of the forming Langmuir monolayer.

Now, having this information on the interaction of citrate-coated IONPs, we pro-ceded to the conjugates IONP@CA_GMP_Tb^3+^. The measurements were performed for two different concentrations of the conjugate in the subphase: 1.4 and 14 mg/L (Figure 9). These higher values as compared to citrate-modified nanoparticles result from the normalization of the mass of the conjugate to correspond to that of IONP_CA added in the preceding experiments presented in Figure 8. Thus, to be able to compare the obtained results for both systems, the concentration of conjugate added to the subphase should be increased.

Isotherms recorded for the DOPC lipid show a systematic increase in surface pressure (Π) with a decreasing surface area per molecule (Apm); see Figure 9. The addition of conjugates with a concentration of 1.4 mg/L reduces the Apm by 5%, while the higher amount of nanocarriers in the subphase (14 mg/L) causes a total decrease of about 10%. The values of the compressibility coefficient for each of the isotherms are about 80–90 mN/m and correspond to the boundary between the liquid-expanded and liquid-condensed phase. These results point out similar but more significant interactions between the DOPC lipid and the conjugate as compared to IONP@CA structures.

For the monolayer made of cardiolipin, the addition of IONP@CA_GMP_Tb3+ conjugates caused a more pronounced influence on the formation of the monolayer depending on the concentration of nanoparticles. The addition of the conjugate at a concentration of 1.4 mg/L causes the expansion of the layer, in which the surface area per one molecule was greater than the monolayer without their addition. The increase in area was 10%. Closer inspection of this monolayer brings forth a small hump at ca. 18 mN/m, leading to even less organized film at higher surface pressures. The 14 mg mg/L nanoconjugates in the subphase resulted in a 13% reduction in the surface area. The calculated compressibility coefficients were respectively 80 mN/m for a monolayer without nanoparticles, 50 mN/m for nanoparticles with a concentration of 1.4 mg/L, and 63 mN/m for nanoparticles with a concentration of 14 mg/L.

On this basis, it can be concluded that the monolayer in the absence of a carrier and in the presence of a significant amount of it (14 mg/L) in the subphase is characterized by the state of the condensed liquid phase. At the same time, the addition of a low concentration of the conjugate to the subphase changes the packing state of the membrane into expanded liquid.

In the case of C_18_NH_2_, Langmuir isotherms show different scenarios compared to the previously discussed behavior of lipid films. The addition of the nanoconjugate to the subphase causes a shift in a curve toward lower values of the surface area, depending on the concentration of IONPs@CA_GMP_Tb^3+^. For a concentration of about 1.4 mg/L, there was a decrease in the surface area ca. 18%, while this change was about 7% for a higher concentration. Additionally, the isotherms show evident phase transitions at both concentrations of nanoconjugates from liquid condensed to liquid-crystalline state at ca. 15 mN/m and 8 mN/m for 1.4 mg/L and 14 mg/L of the conjugates, respectively. This is also evidenced by a steep rise in the compressibility coefficients, reaching up to 300 mN/m and 350 mN/m, respectively, for both concentrations. We think that such behavior can be caused by the reorganization of the octadecylamine layer die to the strong ordering interactions between the Tb^3+^ cation from the nanoconjugate with the free pair of electrons from the nitrogen atom in the small headgroup of octadecylamine.

### 3.3. Magnetic Hyperthermia

The NPs were evenly dispersed and measured in two media: water DI and human serum (HS). The 0.5 mL of the aqueous suspension having a density of 9 mg/mL was inserted into the thermostated copper coil, and the temperature changes were measured with alternating magnetic field in the frequency range from 155 to 633 kHz and with an amplitude from 10 up to 30 kA/m. Measurements were performed to reach the temperature of 55 °C, and the SAR (specific absorption rate) value along with the heating rate of the suspension were measured simultaneously. The most common parameter allowing estimation of the heat conversion efficacy of the sample is SAR, which is defined as the power absorbed per mass of a sample in units of Watt (W) per mass (g) of nanoparticles. Such parameters depend on the magnetic field H (kA/m) and frequency of magnetic field f (kHz), although also the intrinsic loss power ILP parameter is widely used: ILP = SAR/f·H2 to compare the heating of NPs between different experimental setups [39]. As can be seen in Figure 10a,b, the temperature of the suspension of a magnetic bioconjugate in DI water and human serum for 20 kA/m AMF amplitude at various frequencies of alternating magnetic fields increases over time. The fastest rise was generated for the highest frequency of the magnetic field (633 kHz), while a decrease in AMF frequency leads to the decrease in the measured temperature. 

The best suited frequencies to facilitate magnetic hyperthermia are those higher than 330 kHz with an AMF amplitude of at least 20 kA/m. Theoretically, the sample should reach a plateau at the temperature in the range 42–45 °C for magnetic hyperthermia purposes. This is frequently called “mild” hyperthermia, which can effectively stimulate tumor cell apoptosis [40]. Practically, it is performed by switching on and off the magnetic field to maintain the desired temperature for a prolonged time, e.g., 60 min. The AMF field amplitudes and frequencies used in our experiments allow us to reach over 40 °C in less than 5 min. These settings are also below the patient discomfort Brezovich limit (H × f = 5 × 10^9^ A/m·s) [41,42]. The heating rates in DI water and human serum (Figure 10e) are comparable with only a 1.5 °C difference at the end of the measurement. 

The specific absorption rate values obtained for our conjugate IONP@CA_GMP_Tb are shown in Figure 10f. They were calculated using a ZaR subprogram of MaNIaC 1.0 Software (Nanoscale Biomagnetics, Zaragoza, Spain) according to the following equation.
(2)$$\mathrm{SAR}=\frac{d_{l}\cdot {Ce}_{l}}{N_{p}}{\left(\frac{\mathrm{dT}}{\mathrm{dt}}\right)}_{\max}$$
d_l_—dispersant density (kg/m^3^), Ce_l_—dispersant specific heat (kcal/kgoC), N_p_—nanoparticles density (kg/m^3^), T—temperature (oC), and t—time (s).

The SAR values increase from (65.4 ± 1.1) to (99.5 ± 4.5) W/g with the increase in AMF amplitude from 10 to 30 kA/m. The literature reports IONPs with SAR values as high as 3050 W/g for magnetic vortex nanorings [43] through to 1018 W/g for Mn_0.6_Zn_0.4_Fe_2_O_4_ nanospheres [44] and 111 W/g for silica core–shell IONPs [45]. In our previous works regarding magnetic NPs for magnetic hyperthermia, we found SAR values to be in a similar range (105–225 W/g) for lanthanide-doped IONPs [46]. Previously synthesized NPs doped with holmium were characterized by SAR values as high as 350 W/g for trastuzumab-modified SPIONs [47], 400 W/g for core–shell SPIONs trastuzumab bioconjugates [48], and 370 W/g for doxorubicin/epirubicin-modified SPION [49]. The SAR value itself (apart from a dependency on AMF frequency and field amplitude) is connected to the NPs’ size, shape, core modification, ligand attachment, and their amount.

## 4. Discussion

Magnetic iron oxide nanoparticles were obtained with the co-precipitation technique. Unfortunately, bare iron oxide nanoparticles in an aqueous medium tend to agglomerate. Substances such as citric acid are often used to prevent aggregation. It improves the stability of the suspension by repulsive interactions between functional groups of the surfactant. Therefore, the next step of synthesis was the surface coating of the obtained nanoparticles with citric acid as a stabilizer. TEM reveals that nanoparticles have a spherical shape of about 12 nm and are characterized by a low zeta potential value ((−31.1 ± 1.6) mV), which proves the good dispersion properties of the suspension and effective adsorption of citrate anions. The measurements performed by the DLS technique determined the hydrodynamic diameter of the obtained nanoparticles to be (20.9 ± 3.3) nm. The obtained nanoparticles show superparamagnetic properties with a saturation magnetization of 63 emu/g, which is similar to the value in the literature.

Citrate groups on the NPs surface were used as linkers with guanosine-5′-monophosphate and terbium ions, leading to the formation of the magnetic conjugate. The FTIR spectra confirm successful coating with citric acid and GMP along with zeta potential studies. For nanoparticles modified with GMP and Tb^3+^ ions, the zeta potential value is (−15.8 ± 1.0) mV. The increase in its value results from GMP and Tb^3+^ attachment to the negatively charged citrate anions. The conjugate is characterized by the higher value of hydrodynamic diameter (122 ± 10 nm) due to the extensive structure of the attached organic layer and the possibility of creating larger structures.

The analysis of the thermogravimetric curve showing a weight loss of about 50% confirms the presence of additional layers in the conjugate, which is related to the attachment of guanosine monophosphate and lanthanide ions. Additionally, due to the characteristic fluorescent properties of terbium ions, it was also identified using fluorescence spectroscopy.

We also report the effect of the conjugates on synthetic lipid membranes—an equivalent of biological membranes. Langmuir isotherms confirm the slight influence of the nanoparticles modified with citric acid on the structure of biomimetic membranes. In the conjugate with terbium ions, the influence on the way of organizing the lipids in the membrane is related to the functional groups in the lipids and strongly depends on the concentration of the conjugate. The most significant accumulation of nanoconjugate in the membrane occurs for the lipid—cardiolipin.

Hyperthermia study, especially heating profile in human serum, proved that the designed bioconjugate has the potential to be exploited in in vitro experiments—it can reach temperatures over 40 °C in a timely manner (<10 min) with the application of moderate magnetic field frequencies (<400 kHz) and field amplitudes not higher than 20 kA/m. These are the most crucial criteria that need to be met because of equipment capabilities: for flat coils (designed for in vitro study), the frequency cannot exceed 422 kHz and 18 kA/m.

## 5. Conclusions

Our work presents preliminary studies on a conjugate that is meant to make endoradiotherapies possible. The route of synthesis and characterization of magnetic nanostructures modified with terbium ions and an organic layer containing thiopurines analogues are presented. The obtained nanosystem can be considered as a model one, and what is promising is that after two modifications, the substrates can be used in modern treatments of neoplastic diseases, including targeted therapies and magnetic hyperthermia. These modifications should involve replacing the stable terbium ion with ^161^Tb radionuclide, emitting β^−^ radiation, and using the anticancer drug, such as 6-thioguanine, instead of guanine present in GMP. Both modifications will make the conjugate useful in cancer therapies.

## Figures and Tables

**Figure 1 nanomaterials-12-00795-f001:**
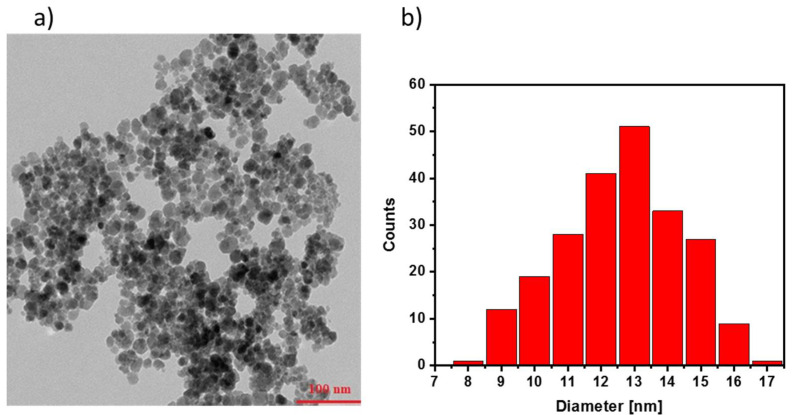
TEM images of IONP@CA (**a**), and size distribution graph (histogram) (**b**) (scale bar = 100 nm).

**Figure 2 nanomaterials-12-00795-f002:**
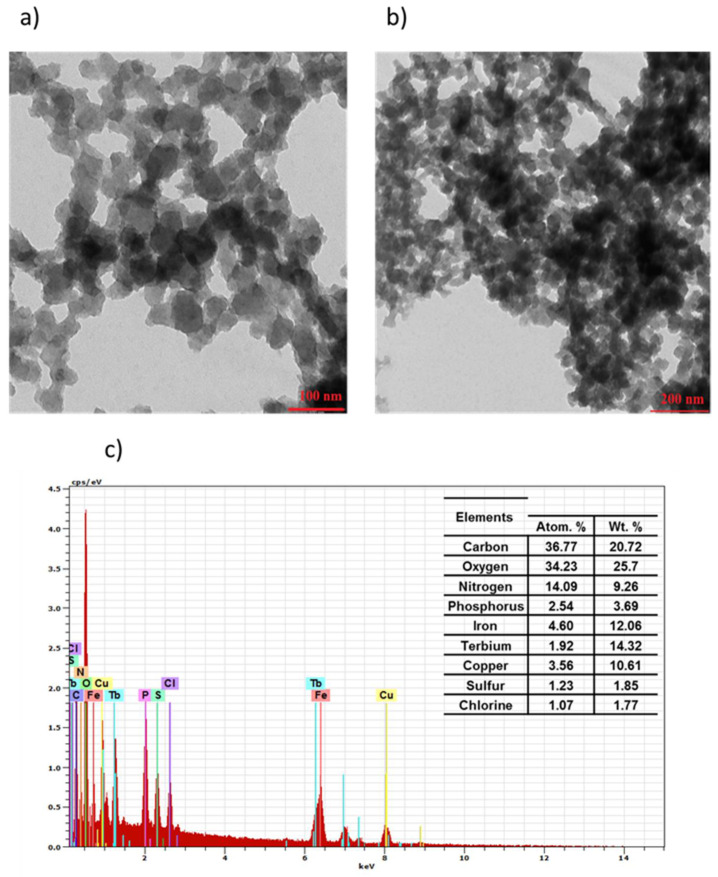
TEM images (**a**,**b**) and EDS spectrum (**c**) of IONPs@CA_GMP_Tb^3+^ (scale bar = 100 nm and 200 nm).

**Figure 3 nanomaterials-12-00795-f003:**
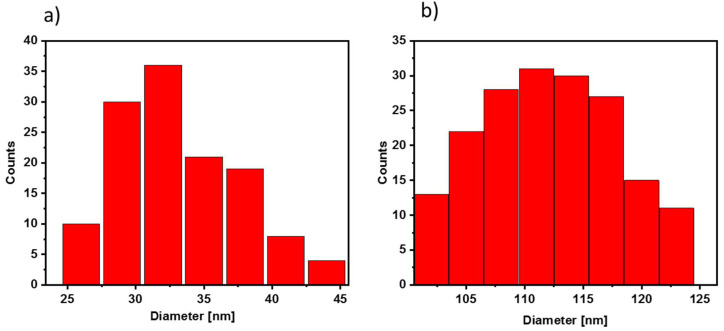
Size distribution graph (data from DLS measurement) of (**a**) IONPs@CA and (**b**) IONPs@CA_GMP_Tb^3+^.

**Figure 4 nanomaterials-12-00795-f004:**
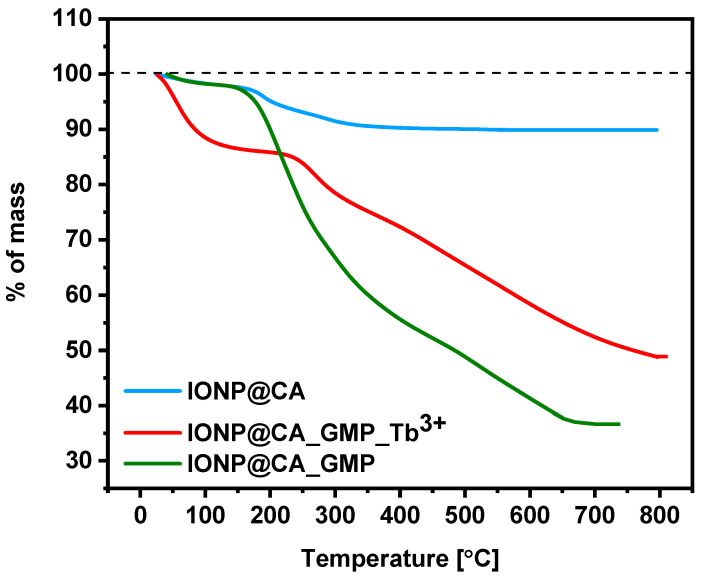
Thermograms of IONPs@CA, IONPs@CA_GMP, and IONPs@CA_GMP_Tb^3+^ (curves marked in the graph).

**Figure 5 nanomaterials-12-00795-f005:**
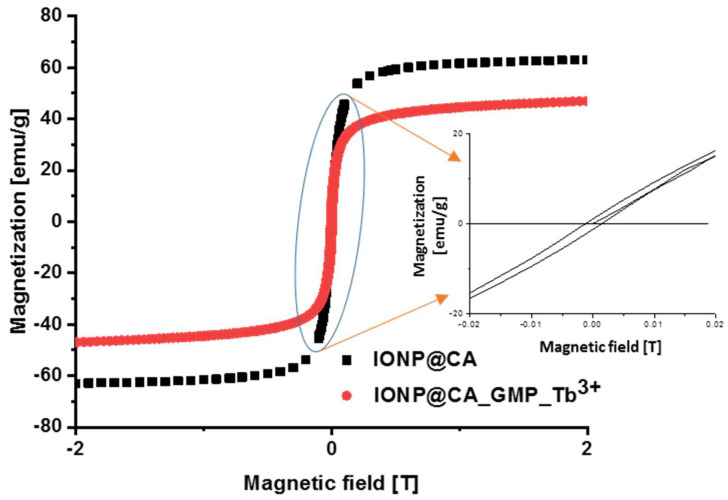
Magnetization measurement for IONPs@CA and IONPs@CA_GMP_Tb^3+^. The inset shows the expansion of the magnetization loop in the range +/− 0.02 T to show the remanence and coercivity of the synthesized nanoparticles.

**Figure 6 nanomaterials-12-00795-f006:**
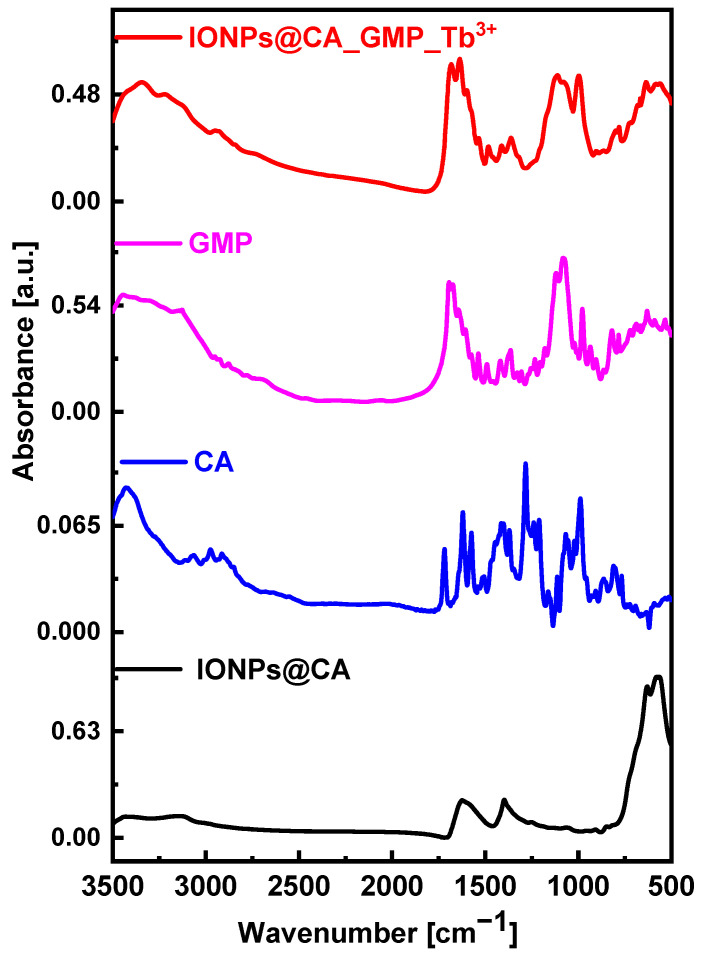
FTIR spectra of: IONPs@CA_GMP_Tb^3+^ (red curve), GMP (magenta curve), CA (blue curve), IONPs@CA (black curve).

**Figure 7 nanomaterials-12-00795-f007:**
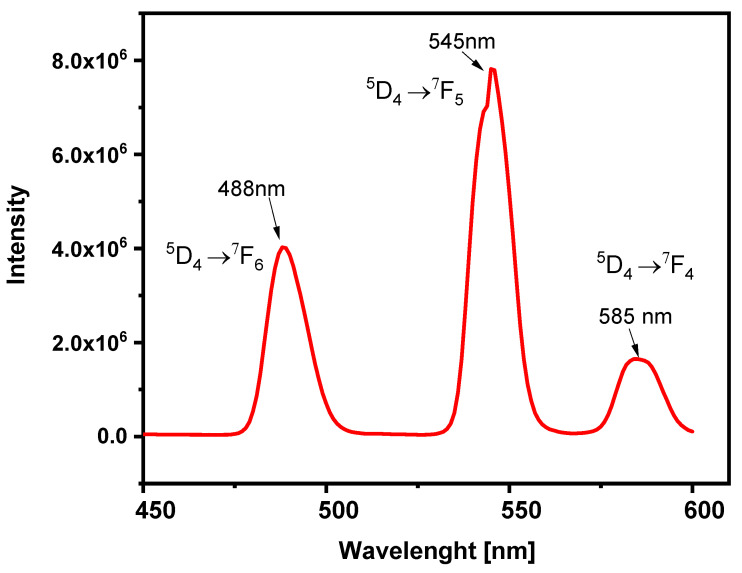
Fluorescence spectrum of IONPs@CA_GMP_Tb^3+^.

**Figure 8 nanomaterials-12-00795-f008:**
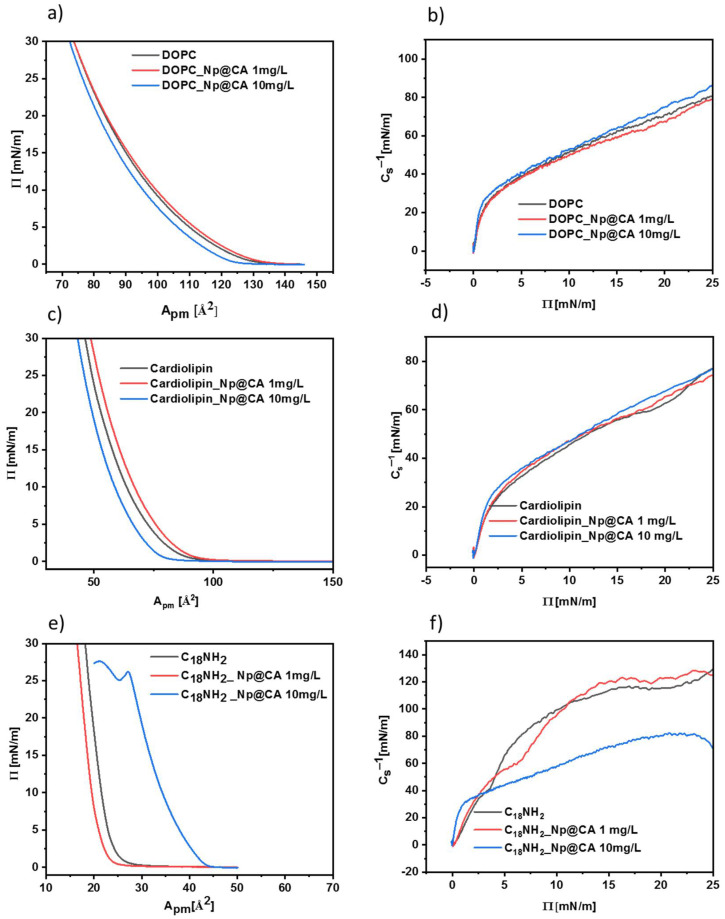
Langmuir isotherm and compressibility coefficient of DOPC (**a**,**b**), C_18_NH_2_ (**c**,**d**), and cardiolipin (**e**,**f**) membranes in the presence and absence of nanoparticles (concentration of Np@CA 1 and 10 mg/L marked in the graph).

**Figure 9 nanomaterials-12-00795-f009:**
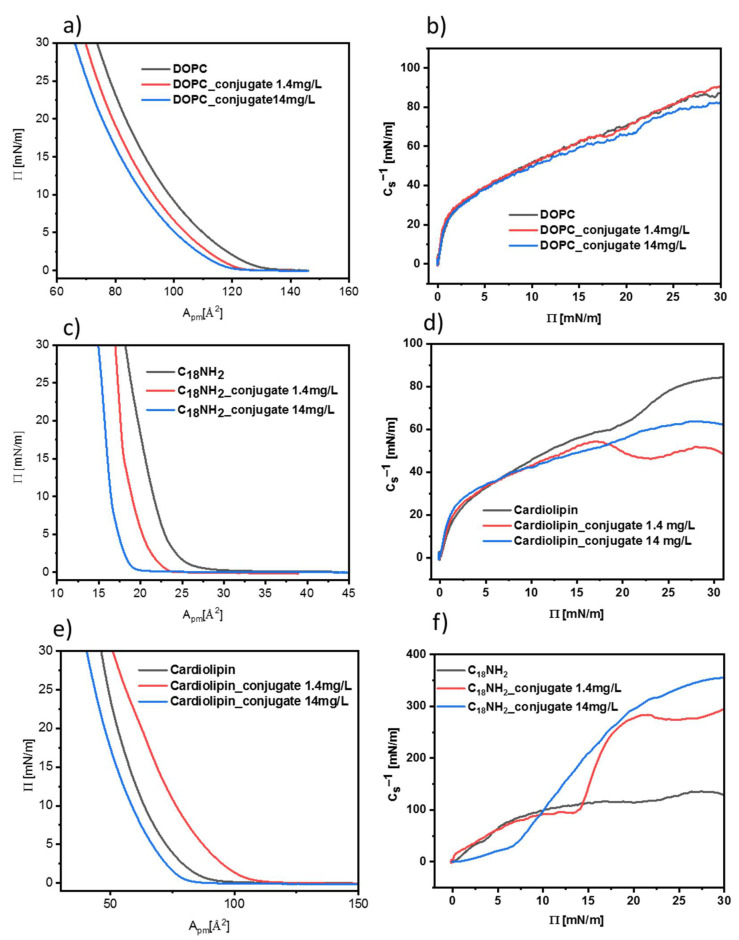
Langmuir isotherm and compressibility coefficient of DOPC (**a**,**b**), C_18_NH_2_ (**c**,**d**), and cardiolipin (**e**,**f**) membranes in the presence and absence of conjugate. (concentration of IONP@CA_GMP_Tb^3+^ 1.4 and 14 mg/L-marked in the graph).

**Figure 10 nanomaterials-12-00795-f010:**
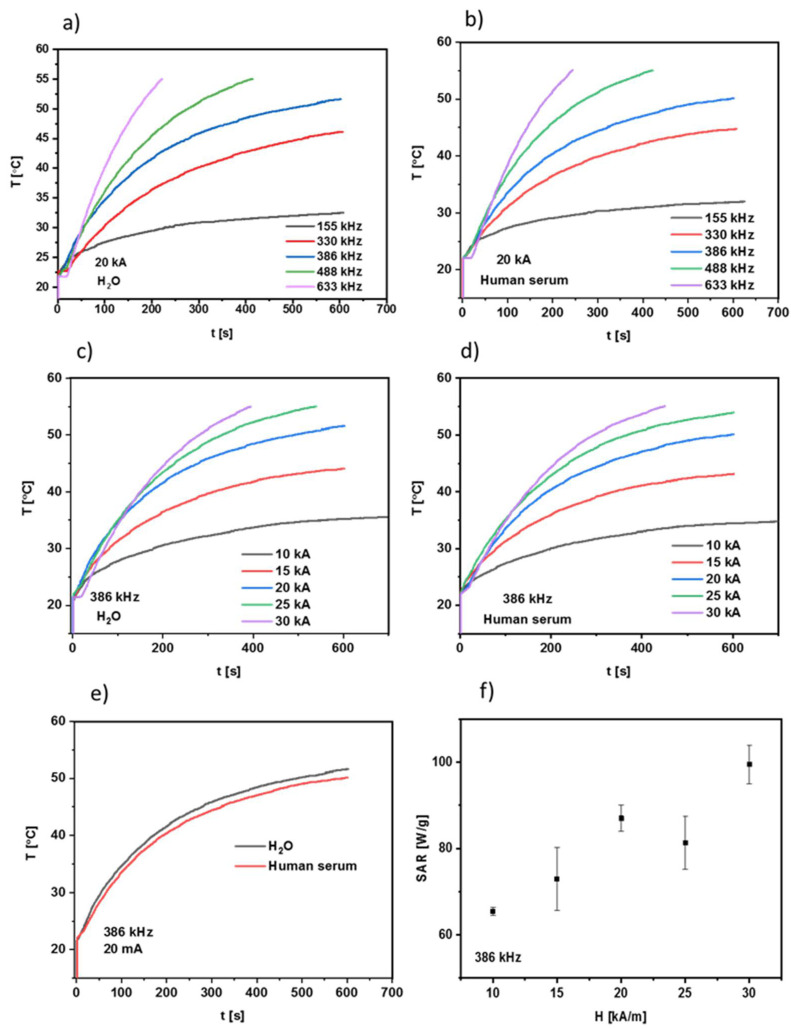
Heating of bioconjugate: (**a**) water DI, 155–633 kHz at 20 kA/m, (**b**) human serum, 155–633 kHz at 20 kA/m, (**c**) water DI, 386 kHz at 10–30 kA/m, (**d**) human serum, 386 kHz at 10–30 kA/m, (**e**) water DI and human serum, 386 kHz at 20 kA/m, (**f**) SAR values for various AMF amplitudes at 386 kHz. Data are expressed as mean ± (n = 3).

## Data Availability

Data can be obtained by contacting authors.

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
