# Peer review of "Synthesis and Characterization of Magnetic Drug Carriers Modified with Tb3+ Ions"

_nanomaterials, 2022, doi:10.3390/nano12050795_

Round 1

Reviewer 1 Report

The aim of this paper is to develop a model for nanoparticles coated with a drug and radioactive isotope. The actual nanoparticles would then be used for tumor treatment.

The article uses moderately good english (there are mistakes, but if the authors do not correct them the MDPI language team should be able to do so) and offers a reasonably clear thread to follow through and understand the purpose of the article.

One thing I think is unfortunate is that the authors used a model instead of the actual nanoparticles they want to use for the application: For one, even though it is maybe not that likely, replacing thioguanosine with guanosine could alter the reactivity under the synthesis conditions, which might result in nanoparticles with different properties. For another, working with radioactive compounds with a short half life requires a much different setup for reactions than stable isotopes, so the synthesis procedure likely could not be immediately transferred to the actual (hopefully) bioactive materials. Furthermore, I expect the guanosine to bind tightly to the IONP thanks to its phosphate group. The way I understand how thioguanidine works is that it is incorporated into DNA, so there should have been tests done to see if thioguanidine can actually be cleaved off the conjugate under physiological conditions, which would not possible with guanosine either.

Another thing, which however does need to be added in my opinion is a cell toxicity test. I understand that this article is more about material synthesis and characterization, but a toxicity test is the least amount of biological testing that should be added to make this manuscript viable. Lanthanides are known to be very toxic, which is for example the reason why gadolinium is generally used as a kinetically very stable complex when injected as a contrast agent. The way that terbium was added here likely forms some kind of complex, but there are no data about its stability, kinetic or thermodynamic (since it isn’t actually known what kind of complex forms). This is why I suggest the authors do comparative cytotoxicity tests between IONP@CA_GMP and IONP@CA_GMP_Tb3+ to see if the Tb ions can actually leak from the material and cause problems.

Mainly because of this I will suggest major revisions to be done. Other more detailed comments follow below:

- l 24: I think after „iron oxides“ you should put a „—„ rather than „ , “

- l 64: you should add a citation for this application of thiopurines

- l 91f: it would be nice if you could write an approximate amount of ammonia solution that you added, the approximate time it took for the addition, and the stirring speed during the addition of ammonia

- a materials section should also be added to list the purity and suppliers of the chemicals used and especially concentration of ammonia

-l 96: what ratio was the water/NH4OH mixture? (probably it would just be better to write this as molarity or percentage of an ammonia solution, since this is what this mixture is)

-l 98f: the three sentences starting from “Unfortunately, bare iron…” belong more into the results/discussion part. I think it would be preferable if you move them there and include a few more sentences about the synthesis to integrate this part better into the results section

-l 105: concentration of the ammonia? was the stirring mechanically or magnetically?

-l 109: what was the amount (mg) of dry nanoparticles that was obtained?

-l 116: how many IONPs were added? (mg)

-l 124f: you should add a few more details about the way the other measurements were conducted in the same way that you did for the VSM measurements; actually I think at least some of the details you already added in the results part, I think it would be better to copy or move them to the methods section

-Fig 1b: from a rough estimation I would say that the addition of the y-values is more than 100%, so I expect the y axis description to be wrong (you probably meant particle count instead of frequency). Please change either the y scale to represent actual frequency (%) or the description of the y axis

-l 171: I think it is better to write EDX or EDS here instead of X ray, to clarify which analysis method is used

-Fig 2a and b: are these from IONPs@CA_GMP or IONPs@CA_GMP_Tb3+? the caption and the description in the text say different things

-l 187: I do not actually see a homogeneous organic shell in the TEM images that could explain such a size increase of 60 nm; it is also unlikely to happen since what you added are monomers, which generally do not have a large size which could form such a large shell. I think it is more likely that there are multiple IONP linked together. Maybe you should rewrite that part of the paragraph (or explain to me more detailed why I am wrong)

-l 207f: I do not think this explanation and the following determination of 10% CA content should be left together like this. This is the first time I read about the explanation that the remains of CA reduce the iron oxide in any article. I have never thought about it like this before, however I agree that this is possible (though it is not the only explanation). Most authors would just mention the 10% as loss of CA without explanation, however, if you do give an explanation like this you should actually do a lengthy calculation to determine the actual CA content because some of the observed weight loss is caused by oxygen bound in iron oxide. In my opinion you can either leave the explanation and do the calculation to determine the actual CA content (which should be somewhere around 9% I guess) or cut the explanation/use a different explanation that does not include reduction of the iron oxides and remain with the 10% CA.

-since you wrote earlier that characterization was done at each step of the synthesis, here I think it would also be interesting to see the TGA curve of IONPs@CA_GMP particles, to see how much Tb actually got adsorbed

-3.1.3 it is nice to have a magnetization curve of the CA coated nanoparticles as comparison, but the actual more important measurement would be the magnetization curve of IONPs@CA_GMP_Tb3+ particles, as those are closer to a potential application

-l 251: ref 29 is in the wrong format

-l 377: should this be IONP@CA instead of IONP?

-l 383: should be “..larger amounts of nanoparticles..” I think

-4. This discussion part to me seems to be like a mixture between discussion and conclusion. While the discussion part can be integrated together with the results, I think I have not seen an article without an explicit conclusion part. I believe this should be added after the discussion. Also, why did you not discuss the hyperthermia experiments here?

-l 507: the radius of Tb is much less than 1 nm..i doubt that contributes this much to the particle size; I think you should consider cutting that part out

Reviewer 2 Report

This paper reports the preparation of iron oxide nanoparticles modified by citric acid, and other molecules combined with Tb3+ to be used in medical purposes.. Then, various ordinary measurements have been done to characterize the property of the products. However, it is not mentioned that the products can be really usable in the real diagnostic and therapeutic systems, because the MNPs forms aggregates in many cases and the size of over 100 nm of the aggregates will be a big problem for the medical use.  The Langmuir layer experiments had no useful results suggesting the utility of the products. Probably, this study is one part of the whole study of the application of radio active Tb3+ MNPs which will be applied to the real tissue or blood. So, the authors should prepare the paper including the MNPs including radio active Tb3+ and the result of its application to the real systems. The report of only the preparation of such MNPs is not new in these days.

In the Langmuir measurements, the equilibrium state must be checked.

Small corrections are required in the manuscript.

Fig. 4 in line 179 should be Fig. 3.

Line 308 requires editing. 

Line 312-322 are almost duplicated in the following sentences.

Reviewer 3 Report

Comments:

In this study, the authors reported the preparation of a magnetic drug carrier modified with terbium (III) ions, which holds potential to achieve targeted anticancer radiation. Detailed characterizations have been caried out to demonstrate the structure and magnetic properties of the conjugates. However, there are still some issues need to be revised. This manuscript can only be accepted in the Nanomaterials after the following issues being well addressed:

  1. What’s the main advantage of the prepared magnetic drug carriers modified with Tb3+ ions compared with other traditional radiotherapeutic drugs?
  2. As the authors proposed that the constructed drug carriers can be served as a promising multifunctional system used in biological imaging, it is suggested to give relevant results in cells.
  3. More ref to be cited:Self-Assembled Nanomaterials for Enhanced Phototherapy of Cancer,ACS Appl. Bio Mater. 2020, 3, 86−106ï¼›Tumor Microenvironment-Specific Functional Nanomaterials for Biomedical Applications Journal of Biomedical Nanotechnology Vol. 16, 1325–1358, 2020

Round 2

Reviewer 1 Report

First I would like to thank the authors for considering my suggestions. Unfortunately I have seen many times that authors were especially reluctant to conduct new experiments which I suggested to them, unlike you. I have also read the other reviewers’ responses. I would like to say that from my point of view the manuscript with the additional experiments is already publishable, and should not be rejected. Nevertheless I still made some comments below, and although the amount of work to be done by you is minor I will select major revisions mainly because I want to see the answers to some of the questions I asked. This includes a question about the two references that one of the reviewers asked to add. While both of them are reviews, in my opinion they do not seem to be as necessary to the manuscript as the other reviewer seems to believe, since they are about loosely related but different subjects from your manuscript. They also happen to be written by the same work group, so I will ask the editor to check if the reviewer belongs to this work group, and if yes, for the editor to suggest you to exclude these specific references. (I might be too suspicious here, but I have seen some reviewers blatantly suggest their own articles to be cited, sometimes as many as 10, which is a practice that I do not agree with). My other comments are added below:

-I am very impressed with the release studies; it looks like the release of GMP is very pH specific, which is a great point for the new material that you made; I am not convinced about GMP binding via N rather than phosphate to the IONP, because all my previous experience says otherwise (phosphate binds and influences formation of IONP greatly when added in the formation reaction, generic iron ammine complexes are either unstable or nonexistent). This is not something you need to do or mention in this manuscript however, but I do feel it might warrant further research in a future article if you can come up with a way to prove it..just something for you to think about for the future.

-from what I know, PBS (phosphate buffered saline) is supposed to have a pH of around 7. I am not sure if you can call a solution of pH 5 still PBS, maybe you should change this formulation or at least give a more detailed description of it (something like mentioning the amounts of NaCl, NaH2PO4, K2HPO4 or whatever salts you used in mmol/l solution)

-Since you added also UV measurements in the supporting information, please write also which device you used there, either in the methods section of the manuscript or in the supporting information. (Did you use 100 µL cuvettes for this by the way or did you dilute the 100 µL aliquots? you do not have to write this in the manuscript, this is just personal curiousity)

-l 253: it should be Figure S1

-I do think a cytotoxicity test is still useful, because your particles will not stay where they are forever, but will eventually be (most likely) be degraded and move into the rest of the body before being excreted, unless you do plan to cut out the tumor surgically as soon as your treatment is done? For that I am happy to see that you included a cytotoxicity study anyway, although I would have preferred to see a comparison between the sample with and without Tb (but you don’t have to do that now, the study you did is enough for me). However, you should also mention this study in the main manuscript, at least in 1-2 sentences. From my discussion with a colleague who does cytotoxicity tests I learned that the results are pretty encouraging so far.

-just a side note, not to be included in the manuscript: Under inert conditions citric acid does not necessarily convert into carbon dioxide and remaining carbon.

https://doi.org/10.1016/0040-6031(86)87081-2

As this paper suggests, it can form aconitic acid first, which can then decompose to acetone (which is volatile, although the stoichiometry in the paper does not seem to me correct, but even isopropanol would be volatile), and not leaving any carbon residue. Of course, the iron oxide in your case can play the role of a catalyst, which might make entirely different reactions also possible. It isn’t really important what is formed for the manuscript though, I just wanted to point out some alternatives that could also happen to the citric acid coating, which is why I suggested in my first review that it would be better to just leave this part of the discussion out.

The TGA curve recorded  for nanoparticles modified  with CA and GMP shows a

decrease to a value of about 40%. This is confirmed by the 60% content of the organic shell (GMP

and CA). Based on both TGA curves (with and without terbium ions), the terbium content was

estimated at 16.2% (assuming additivity in both syntheses)

-since you measured it already, could you include this curve together with this explanation in the manuscript please, or if you do not like that, in the supporting information at least (with a reference from the manuscript towards the supporting information figure)? I think this is interesting enough to be mentioned somewhere.

Reviewer 2 Report

In this study, the stability of the chemical binding of Tb(III) is crucially important. This point should be proved experimentally. 

Round 3

Reviewer 1 Report

I would like to thank the authors very much for their interesting responses. I enjoyed reading them and getting a bit deeper understanding of the research matter. I think the manuscript can be accepted in its present form now.

(one small comment: I did not consider the binding of GMP via N to the citrate carboxylate groups, thanks for pointing that out. I still would have expected that a kind of ligand exchange happens, because I would have thought that the phosphate binds more strongly to iron than citrate, but maybe it is not so. It is definitely not impossible, since citrate is a strong complexing agent. My error of thinking was basically to consider the citrate coating easily replacable.)